# Design, Synthesis and Pharmacological Evaluation of Novel Conformationally Restricted *N*-arylpiperazine Derivatives Characterized as D_2_/D_3_ Receptor Ligands, Candidates for the Treatment of Neurodegenerative Diseases

**DOI:** 10.3390/biom12081112

**Published:** 2022-08-12

**Authors:** Thayssa Tavares da Silva Cunha, Rafaela Ribeiro Silva, Daniel Alencar Rodrigues, Pedro de Sena Murteira Pinheiro, Thales Kronenberger, Carlos Maurício R. Sant’Anna, François Noël, Carlos Alberto Manssour Fraga

**Affiliations:** 1Laboratório de Avaliação e Síntese de Substâncias Bioativas (LASSBio), Instituto de Ciências Biomédicas, Universidade Federal do Rio de Janeiro, Rio de Janeiro 21941-902, RJ, Brazil; 2Programa de Pós-Graduação em Farmacologia e Química Medicinal, Instituto de Ciências Biomédicas, Universidade Federal do Rio de Janeiro, Rio de Janeiro 21941-902, RJ, Brazil; 3Laboratório de Farmacologia Bioquímica e Molecular, Instituto de Ciências Biomédicas, Universidade Federal do Rio de Janeiro, Rio de Janeiro 21941-902, RJ, Brazil; 4Department of Medical Oncology and Pneumology, Internal Medicine VIII, University Hospital of Tübingen, Otfried-Müller-Strasse, 72076 Tübingen, Germany; 5Instituto de Química, Universidade Federal Rural do Rio de janeiro (UFRRJ), Rio de Janeiro 23890-000, RJ, Brazil

**Keywords:** dopamine receptors, *N*-arylpiperazine, neurodegenerative diseases, binding, 1,3-benzodioxole, sulfonamide

## Abstract

Most neurodegenerative diseases are multifactorial, and the discovery of several molecular mechanisms related to their pathogenesis is constantly advancing. Dopamine and dopaminergic receptor subtypes are involved in the pathophysiology of several neurological disorders, such as schizophrenia, depression and drug addiction. For this reason, the dopaminergic system and dopamine receptor ligands play a key role in the treatment of such disorders. In this context, a novel series of conformationally restricted *N*-arylpiperazine derivatives (**5a**–**f**) with a good affinity for D_2_/D_3_ dopamine receptors is reported herein. Compounds were designed as interphenylene analogs of the drugs aripiprazole (2) and cariprazine (3), presenting a 1,3-benzodioxolyl subunit as a ligand of the secondary binding site of these receptors. The six new *N*-arylpiperazine compounds were synthesized in good yields by using classical methodologies, and binding and guanosine triphosphate (GTP)-shift studies were performed. Affinity values below 1 μM for both target receptors and distinct profiles of intrinsic efficacy were found. Docking studies revealed that Compounds **5a**–**f** present a different binding mode with dopamine D_2_ and D_3_ receptors, mainly as a consequence of the conformational restriction imposed on the flexible spacer groups of 2 and 3.

## 1. Introduction

Dopamine is a key neurotransmitter involved in several physiological processes for the full functioning of the body, such as voluntary movements, affection, sleep, attention, memory, learning, hormonal regulation and cardiovascular and immune functions [1,2,3]. The degeneration of dopaminergic neurons in the substantia nigra causes an inhibition of dopaminergic signaling, which can generate rigor, tremor, bradykinesia and postural instability, the main symptoms of Parkinson’s disease (PD) [4,5,6]. However, the mesolimbic pathway is directly involved with the mechanisms of emotion control and reward. Thus, the alteration of dopaminergic neurotransmission in this area is related to the pathophysiology of several diseases, such as schizophrenia and drug addiction [7,8,9,10,11,12]. In addition, other pathophysiological processes are also involved in the alteration of dopaminergic neurotransmission, such as the establishment of arterial hypertension, bipolar disorder and major depression [13,14,15,16].

The dopaminergic system comprises five receptor subtypes divided into two families: D_1_-like (D_1_; D_5_) and D_2_-like (D_2_; D_3_; D_4_). Such subdivision is based on structural differences, such as the homology between their amino acid sequences, as well as their molecular actions, resulting from their different cell signaling processes [1,17,18].

The importance of dopaminergic pathways and receptor modulators in the control of neurodegenerative diseases led to the development of drugs such as the classical typical antipsychotic haloperidol (**1**), a D_2_ receptor antagonist (Ki = 0.89 nM) [19,20], and the atypical antipsychotics aripiprazole (**2**) (Ki D_2_ = 0.34 nM; Ki D_3_ = 0.8 nM) and cariprazine (**3**) (Ki D_3_ = 0.085 nM; Ki D_2_ = 0.49 nM), as partial agonists of D_2_ and D_3_ receptors [21,22,23,24,25,26], approved for the treatment of schizophrenia and bipolar disorder (Figure 1).

Other analog *N*-phenylpiperazine compounds, now showing intrinsic efficacy as antagonists, have also been developed to act in the treatment of dependence and drug addiction [27,28,29]. Among these compounds, compound (**4**) (Ki D_3_ = 0.118 nM; Ki D_2_ = 12.9 nM), a D_2_/D_3_ receptor antagonist developed by Boateng et al. (2015), guards, in its chemical structure, important similarities with cariprazine (**3**), in the presence of an arylpiperazine subunit (also present in aripiprazole) and an amide group. However, compound (**4**) also presents differences such as the introduction of an alkyl spacer that gives greater conformational freedom to the compound in relation to cariprazine (**3**). Furthermore, the presence of an indole subunit appears to be responsible for the change in intrinsic efficacy from a partial agonist in cariprazine (**3**) to an antagonist in derivative (**4**) (Figure 1) [30].

In this context, considering the multifactorial behavior of these neurodegenerative diseases and the importance of finding novel compounds that combine the structural requirements of just one molecule to act on dopamine D_2_/D_3_ receptors with a fine-tuning adjustment of intrinsic efficacies, we reported herein a new series of conformationally restricted *N*-arylpiperazine derivatives (**5a**–**f**) presenting moderate affinity for D_2_/D_3_ dopamine receptors. The compounds were designed as interphenylene analogs of the drugs aripiprazole (**2**) and cariprazine (**3**), having a 1,3-benzodioxole subunit (A) as a ligand for the secondary binding site of these receptors. The pharmacophoric arylpiperazine subunit was preserved in the design concept of the new series of derivatives (**5a**–**f**) (Figure 1). Replacement of the alkyl spacer with an interphenylene spacer was proposed, bringing a conformational restriction [31]. In addition, the amide group present in the amide derivative (**4**) was replaced by a sulfonamide, which has additional points capable of interacting with the bioreceptor [32]. Aromatic substituents such as phenyl, 2-methoxyphenyl and 2,3-dichlorophenyl at position 4 of the piperazine ring were used to evaluate the ortho effect on the coplanarity between the piperazine ring and the aromatic ring [33]. The 1,3-benzodioxole subunit A (Figure 1) attached to a sulfonamide group was chosen due to isosteric relationships with the indoleamide subunit present in compound (**4**), as previously described by our laboratory [34].

Classic synthetic methodologies, molecular modeling studies, and binding and GTP-shift experiments were performed. Six new *N*-phenylpiperazine derivatives (**5a**–**e**) were obtained, with affinity values below 1 μM and distinct profiles of intrinsic efficacy (Figure 1).

## 2. Materials and Methods

### 2.1. Chemistry

All commercially available reagents and solvents were used without further purification. Reactions were routinely monitored by thin-layer chromatography (TLC) on silica gel (F245 Merck plates), and the products were visualized with an ultraviolet (UV) lamp (254 and 365 nm). ^1^H and ^13^C nuclear magnetic resonance (NMR) spectra were determined in dimethyl sulfoxide (DMSO)-d6 solutions using a VARIAN 500-MR spectrometer (Varian, Palo Alto, CA, USA) operating at 500 and 125 MHz, respectively. The chemical shifts are given in parts per million (δ) from solvent residual peaks, and the coupling constant values (J) are given in Hz. Signal multiplicities are represented by s (singlet), d (doublet), dd (double doublet), t (triplet), m (multiplet) and br (broad signal).

Infrared spectra were obtained using a Thermo Nicolet Avatar 330 FTIR (Thermo Fisher Scientific, Waltham, MA, USA) spectrometer equipped with a smart endurance diamond ATR unit for direct measurements. The melting points (MPs) were determined on a Quimis Model Q340.23 apparatus in triplicate.

Microanalyses were carried out using a Thermo Scientific Flash EA 1112 series CHN-Analyzer, using a Mettler MX5 electronic balance.

The purity of the synthesized compounds was determined by high-performance liquid chromatography (HPLC), which was performed in a Shimadzu LC20AD apparatus (Shimadzu, Tokyo, Japan) using a Kromasil 100–5C18 column (4.6 mm × 250 mm) (Kromasil, Bohus, Sweden) and an SPD-M20A detector (Diode Array) at wavelengths ranging from 238 to 287 nm for analyte quantification and a constant flow rate of 1 mL/min. The automatic injector was programmed so that the volume of sample injected per analysis corresponded to 20 μL. The mobile phases used were 60% acetonitrile and 40% water, 60% methanol and 40% water, and 80% ethanol and 20% water; the pH of the mobile phase was adjusted to 3 and 6.5 according to the type of compound to be analyzed. The solvents used for HPLC-PDA analysis were HPLC purity grade (Tedia^®^).

#### 2.1.1. General Procedure for the Synthesis of 4-Nitrobenzyl-phenylpiperazine Intermediates **8a**–**c**

In a 125-mL flask, 0.151 g (1 mmol) of 4-nitrobenzaldehyde (**10**) was dissolved in 30 mL of absolute ethanol. Then, 1 equivalent of the respective phenylpiperazines (**9a**–**c**) and 0.5 equivalents of zinc chloride (ZnCl_2_; 0.07 g; 0.5 mmol) were added to the solution. The reaction mixture was kept under constant stirring at 60 °C.

After 2 h, 3.2 equivalents of sodium cyanoborohydride (NaBH_3_CN; 0.2 g; 3.2 mmol) was added in 2 portions every 1 h. The complete consumption of starting materials was evidenced 24 h after the addition of the reducing agent by TLC using a mixture of hexane:ethyl acetate (60:40) as the eluent.

The isolation of the product was carried out by extraction in a separatory funnel using ethyl acetate and saturated sodium bicarbonate solution. The organic phase was separated, dried with anhydrous sodium sulfate and concentrated at reduced pressure, furnishing the desired 4-nitrobenzyl-phenylpiperazines (**8a**–**c**), as described next.

##### 1-(4-Nitrobenzyl)-4-phenylpiperazine (**8a**)

Intermediate **8a** was obtained with 73% yield as a yellow crystalline solid and had a melting point of 123–126 °C. ^1^H NMR (500 MHz, DMSO-*d6*) δ (ppm): 8.21 (2H, d, J = 8.8 Hz, H14, H16), 7.62 (2H, d, J = 8.8 Hz, H13, H17), 7.20 (2H, dd, J1 = 8.7 Hz, J2 = 7.3 Hz, H2, H6), 6.91 (2H, d, J = 7.9 Hz, H3, H5), 6.76 (1H, t, J = 7.3 Hz, H1); 3.66 (2H, s, H11), 3.14 (4H, m, H7, H10), 2.53 (4H, m, H8, H9); IR (ATR, cm^−1^): 1511 and 1345 (*v*-NO_2_).

##### 1-(4-Nitrobenzyl)-4-phenylpiperazine (**8b**)

Intermediate **8b** was obtained with 70% yield as a yellow crystalline solid and had a melting point of 106–109 °C. ^1^H NMR (500 MHz, DMSO-*d6*) δ (ppm): 8.20 (2H, d, J = 8.7 Hz, H14, H16), 7.62 (2H, d, J = 8.7 Hz, H13, H17), 6.93 (2H, m, H2, H6), 6.86 (2H, m, H5, H1), 3.75 (3H, s, H18), 3.66 (2H, s, H11), 2.97 (4H, br, H7, H10), 2.53 (4H, br, H8, H9); IR (ATR, cm^−1^): 1497 and 1347 (*v*-NO_2_).

##### 1-(4-Nitrobenzyl)-4-phenylpiperazine (**8c**)

Intermediate **8c** was obtained with 63% yield as a yellow crystalline solid and had a melting point of 131–133 °C. ^1^H NMR (500 MHz, DMSO-*d6*) δ (ppm): 8.20 (2H, m, H14, H16), 7.62 (2H, d, J = 8.7 Hz, H13, H17), 7.28 (2H, dd, J1 = 7.5 Hz, J2 = 5.3 Hz, H2, H6), 7.14 (1H, dd, J1 = 6.8 Hz, J2 = 2.8 Hz, H1), 3.68 (2H, s, H11), 2.99 (4H, br, H7, H10), 2.56 (4H, br, H8, H9); IR (ATR, cm^−1^): 1519 and 1341 (*v*-NO_2_).

#### 2.1.2. General Procedure for the Synthesis of Key Intermediates 4-((4-Phenylpiperazin-1-yl)methyl)anilines (**6a**–**c**)

In a 50 mL flask containing a mixture of EtOH:H_2_O (10:10 mL), 0.15 g of the respective 1-(4-nitrobenzyl)-4-phenylpiperazines (**8a**–**c**) was added along with 3 equivalents of metallic iron (Fe^0^) and 5 equivalents of ammonium chloride (NH_4_Cl). The obtained mixture was refluxed at 80 °C with constant stirring.

The progress of the reaction was monitored by TLC using hexane:ethyl acetate (60:40) as the eluent. The end of the reaction was observed after 1 h. Then, the reaction mixture was filtered out hot through Celite, and isolation was carried out by extraction in a separatory funnel using ethyl acetate and saturated sodium bicarbonate solution. The organic phase was dried with anhydrous sodium sulfate and filtered and concentrated under reduced pressure, furnishing the desired 4-((4-phenylpiperazin-1-yl)methyl)anilines (**6a**–**c**), as described next.

##### 4-((4-Phenylpiperazin-1-yl)methyl)aniline (**6a**)

Intermediate **6a** was obtained in 96% yield as a yellowish oil. ^1^H NMR (500 MHz, DMSO-*d6*) δ (ppm): 7.18 (2H, m, H13, H17), 6.95 (2H, d, J = 8.3 Hz, H2, H6), 6.90 (2H, d, J = 7.9 Hz, H3, H5), 6.75 (1H, t, J = 7.3 Hz, H1), 6.52 (2H, d, J = 8.3 Hz, H14, H16), 4.94 (2H, s, H18), 3.31 (2H, s, H11), 3.09 (4H, m, H7, H10), 2.45 (4H, m, H8, H9); IR (ATR, cm^−1^): 3448 and 3356 (*v*-NH_2_).

##### 4-((4-Phenylpiperazin-1-yl)methyl)aniline (**6b**)

Intermediate **6b** was obtained with 82% yield as a yellowish oil. ^1^H NMR (500 MHz, DMSO-*d6*) δ (ppm): 6.98 (2H, d, J = 8.2 Hz, H13, H17), 6.92 (2H, m, H2, H6), 6.85 (2H, m, H1, H5), 6.53 (2H, m, H14, H16), 3.75 (3H, s, H18), 2.96 (4H, br, H7, H10), 2.55 (4H, br, H8, H9); IR (ATR, cm^−1^): 3332 and 3213 (*v*-NH_2_).

##### 4-((4-Phenylpiperazin-1-yl)methyl)aniline (**6c**)

Intermediate **6c** was obtained with 60% yield as a yellowish oil. ^1^H NMR (500 MHz, DMSO-*d6*) δ (ppm): 7.28 (2H, m, H13, H17), 7.12 (1H, dd, J1 = 6.8 Hz, J2 = 2.9 Hz, H1), 6.94 (2H, d, J = 8.3 Hz, H5, H6), 6.51 (2H, m, H14, H16), 4.95 (2H, br, H18), 3.35 (2H, br, H11), 2.95 (4H, br, H7, H10); IR (ATR, cm^−1^): 3454 and 3361 (*v*-NH_2_).

#### 2.1.3. General Procedure for the Synthesis of Potassium Methylenedioxybenzenesulfonates (**12a**–**b**)

A solution of 0.1 g of the benzodioxoles (**11a**–**b**) and 3 equivalents of acetic anhydride in 2.2 mL of ethyl acetate was cooled to 0 °C. Then, a solution containing 1.3 equivalents of concentrated sulfuric acid (d = 1.84) in 0.3 mL of ice-cold ethyl acetate was added dropwise over 5 min. The mixture was stirred for 2 h while warming to room temperature. After this time, a solution containing 1.5 equivalents of potassium acetate in 0.56 mL of 95% ethanol was added dropwise with stirring. After 30 min, the potassium salts (**12a**–**b**) were isolated by filtration under reduced pressure.

##### Benzo [d] [1,3] Dioxole-5-potassium Sulfonate (**12a**)

Salt **12a** was obtained in 89% yield as a white solid, with a melting point of 172–174 °C; IR (ATR, cm^−1^): 1310 and 1163 (ʋ S=O).

##### 6-Methylbenzo [d] [1,3] Dioxole-5-potassium Sulfonate (**12b**)

Salt **12b** was obtained in 93% yield as a white solid, with a melting point between 187–189 °C; IR (ATR, cm^−1^): 1345 and 1179 (*v* S=O), 651 (*v* S-O).

#### 2.1.4. General Procedure for the Synthesis of Methylenedioxy-Benzenesulfonyl Chlorides (**7a**–**b**)

To 0.15 g of the respective potassium salts (**12a**–**b**), a solution containing 5.4 equivalents of thionyl chloride containing a catalytic amount of anhydrous N,N-dimethylformamide (DMF) was quickly added. The resulting mixture was stirred at 60 °C for 4 h. At the end of this time, a sufficient amount of crushed ice was added to the mixture, resulting in the formation of a precipitate, the corresponding sulfonyl chlorides (**7a**–**b**), which were then collected by filtration under reduced pressure.

##### Benzo[d] [1,3] Dioxole-5-sulfonyl Chloride (**7a**)

Sulfonyl chloride **7a** was obtained in 76% yield as a yellowish solid, with a melting point of 45–48 °C. ^1^H NMR (500 MHz, DMSO-*d6*) δ (ppm): 7.14 (1H, dd, J1 = 8 Hz, J2 = 1.6 Hz, H2), 7.05 (1H, d, J = 1.6 Hz, H6), 6.84 (1H, dd, J1 = 7.6 Hz, J2 = 3.7 Hz, H5), 6.01 (2H, s, H7); IR (ATR, cm^−1^): 1373 and 1159 (*v* S=O).

##### 6-Methylbenzo [d] [1,3] Dioxole-5-sulfonyl Chloride (**7b**)

Sulfonyl chloride **7b** was obtained in 95% yield as a yellowish solid, with a melting point between 75–78 °C. ^1^H NMR (500 MHz, DMSO-*d6*) δ (ppm): 7.21 (1H, s, H2), 6.71 (1H, s, H5); 5.95 (2H, s, H7), 2.42 (3H, s, H8); IR (ATR, cm^−1^): 1375 and 1181 (*v* S=O).

#### 2.1.5. General Procedure for the Synthesis of 1,3-Benzodioxolyl-Sulfonamide *N*-arylpiperazine Derivatives **5a**–**f**

In a G30-type microwave tube containing a solution with 0.08 g of the respective 4-((4-phenylpiperazin-1-yl)methyl)anilines (**6a**–**c**) and 10 mL of ethanol, 1 equivalent of the desired sulfonyl chloride (**7a** or **7b**) was added. The reaction mixture was irradiated in a microwave oven for 30 min at 100 °C. For the isolation of the products, the reaction medium was concentrated and extracted in a separatory funnel using ethyl acetate and water at pH = 10 (adjusted with 10% NaOH solution). The organic phase was dried with anhydrous sodium sulfate, and the solvent was evaporated under reduced pressure. The obtained residue was submitted to a purification step by silica gel column chromatography using hexane/ethyl acetate as the mobile phase in a gradient (90:10 to 70:30). Target compounds **5a**–**f** were obtained in moderate to good yields, as described below.

##### 6-Methyl-*N*-(4-((4-phenylpiperazin-1-yl)methyl)phenyl) Benzo [d] [1,3] Dioxole-5-sulfonamide (**5a**)

*N*-Phenylpiperazine derivative **5a** was obtained in 36% yield as a white solid, with a melting point of 154–156 °C. ^1^H NMR (500 MHz, DMSO-*d6*) δ (ppm): 10.26 (1H, s, H18), 7.33 (1H, s, H20), 7.17 (4H, dd, J1 = 12.6 Hz, J2 = 5.7 Hz, H2, H6, H14, H16), 7.02 (2H, d, J = 8.4 Hz, H13, H17), 6.92 (1H, s, H24), 6.88 (2H, d, J = 7.2 Hz, H3, H5), 6.75 (1H, t, J = 7.2 Hz, H1), 6.07 (2H, s, H22), 3.38 (2H, s, H11), 3.07 (4H, m, H8, H9), 2.47 (3H, s, H26), 2.42 (4H, m, H7, H10); ^13^C NMR (500 MHz, DMSO-*d6*) δ (ppm): 151.05 (C4), 150.69 (C23), 145.31 (C21), 136.48 (C15), 132.78 (C19), 130.42 (C12), 129.82 (C25), 128.95–109.27 (C1, C2, C3, C5, C6, C13, C14, C16, C17, C20, C24), 102.34 (C22), 61.43 (C11), 52.52 (C8, C9), 48.19 (C7, C10), 19.67 (C26); IR (ATR, cm^−1^): 1308 and 1147 (ʋ S=O), 3256 and 1577 (*v* N-H). Purity (HPLC): 99.0%; Anal. calcd. for C25H27N3O4S: C, 64.50; H, 5.85; N, 9.03; Found: C, 64.61; H, 5.83; N, 8.99.

##### *N*-(4-((4-(2-Methoxyphenyl)piperazin-1-yl)methyl)phenyl)-6-methylbenzo [d] [1,3] Dioxole-5-sulfonamide (**5b**)

*N*-Phenylpiperazine derivative **5b** was obtained in 85% yield as a yellowish solid, with a melting point of 183–185 °C. ^1^H NMR (500 MHz, DMSO-*d6*) δ (ppm): 10.26 (1H, s, H18), 7.33 (1H, s, H20), 7.16 (2H, d, J = 8.4 Hz, H14, H16), 7.02 (2H, d, J= 8.4 Hz, H13, H17); 6.91 (3H, dd, J1 = 13.4 Hz, J2 = 6.3 Hz, H2, H6, H24), 6.84 (2H, d, J = 3.7 Hz, H1, H3), 6.08 (2H, s, H22), 3.74 (3H, s, H27), 3.38 (2H, s, H11), 2.91 (4H, br, H8, H9), 2.47 (3H, s, H26), 2.43 (4H, br, H7, H10); ^13^C NMR (500 MHz, DMSO-*d6*) δ (ppm): 151.96 (C5), 150.58 (C23), 145.23 (C21), 141.23 (C4), 136.37 (C15), 132.68 (C19), 130.44 (C12), 129.72 (C25), 122.29–109.13 (C1, C2, C3, C6, C13, C14, C16, C17, C20, C24), 102.2 (C22), 61.47 (C11), 55.27 (C8), C9), 49.98 (C7, C10), 19.57 (C26); IR (ATR, cm^−1^): 1323 and 1150 (ʋ S=O), 3259 and 1593 (ʋ N-H). Purity (HPLC): 97.0%; Anal. calcd. for C26H29N3O5S: C, 63.01; H, 5.90; N, 8.48; Found: C, 63.16; H, 5.89; N, 8.46. 

##### *N*-(4-((4-(2,3-Dichlorophenyl)piperazin-1-yl)methyl)phenyl)-6-methylbenzo [d] [1,3] Dioxole-5-sulfonamide (**5c**)

*N*-Phenylpiperazine derivative **5c** was obtained in 21% yield as a white solid, with a melting point of 192–194 °C. ^1^H NMR (500 MHz, DMSO-*d6*) δ (ppm): 10.25 (1H, s, H18), 7.33 (1H, s, H20), 7.27 (2H, m, H14, H16), 7.17 (2H, d, J = 8.5 Hz, H13, H17), 7.10 (1H, dd, J1 = 6.6 Hz, J2 = 3.0 Hz, H24), 7.02 (2H, d, J = 8.5 Hz, H2, H3), 6.92 (1H, s, H1), 6.07 (2H, s, H22); 3.40 (2H, s, H11), 2.94 (4H, br, H8, H9), 2.47 (4H, br, H7, H10); ^13^C NMR (500 MHz, DMSO-*d6*) δ (ppm): 151.24 (C4), 150.73 (C23), 145.35 (C21), 136.53 (C15), 133.23 (C19), 132.85 (C6), 132.69 (C12), 130.45 (C25), 129.94 (C5), 128.54–109.30 (C1, C2, C3, C13, C14, C16, C17, C20, C24), 102.37 (C22), 61.42 (C11), 52.61 (C8, C9), 50.96 (C7, C10), 19.71 (C26); IR (ATR, cm^−1^): 1328 and 1151 (ʋ S=O), 3286 and 1573 (ʋ N-H). Purity (HPLC): 96.0%; Anal. calcd. for C25H25Cl2N3O4S: C, 56.18; H, 4.71; N, 7.86; Found: C, 56.34; H, 4.69; N, 7.83. 

##### *N*-(4-((4-Phenylpiperazin-1-yl) Methyl) Phenyl) Benzo [d] [1,3] Dioxole-5-Sulfonamide (**5d**)

*N*-Phenylpiperazine derivative **5d** was obtained in 56% yield as a yellowish solid, with a melting point of 178–180 °C. ^1^H NMR (500 MHz, DMSO-*d6*) δ (ppm): 10.12 (1H, br, H18), 7.29 (1H, dd, J1 = 8.2 Hz, J2 = 1.9 Hz, H20), 7.19 (2H, d, J = 1.9 Hz, H14, H16), 7.17 (3H, d, J = 11.1 Hz, H2, H6, H25), 7.05 (2H, m H13, H17), 7.00 (1H, d, J = 8.2 Hz, H24), 6.88 (2H, dd, J1 = 8.7 Hz, J2 = 0.8 Hz, H3, H5), 6.75 (1H, t, J = 7.3 Hz, H1), 6.11 (2H, s, H22), 3.41 (2H, s, H11), 3.08 (4H, m, H7, H10), 2.44 (4H, m, H8, H9); ^13^C NMR (500 MHz, DMSO-*d6*) δ (ppm): 151.07 (C4), 147.86 (C23), 136.65 (C21), 132.89 (C15), 129.90 (C12), 129.01 (C19), 122.59–106.50 (C1, C2, C3, C5, C6, C13, C14, C16, C17, C20, C24), 102.62 (C22), 61.43 (C11), 52.53 (C8, C9), 48.20 (C7, C10); IR (ATR, cm^−1^): 1326 and 1142 (ʋ S=O), 3273 and 1577 (ʋ N-H). Purity (HPLC): 95.0%; Anal. calcd. for C24H25N3O4S: C, 63.84; H, 5.58; N, 9.31; Found: C, 64.01; H, 5.57; N, 9.33. 

##### *N*-(4-((4-(2-Methoxyphenyl)piperazin-1-yl)methyl)phenyl) Benzo [d] [1,3] Dioxole-5-sulfonamide (**5e**)

*N*-Phenylpiperazine derivative **5e** was obtained in 21% yield as a yellowish solid, with a melting point of 149–151 °C. ^1^H NMR (500 MHz, DMSO-*d6*) δ (ppm): 10.11 (1H, br, H18), 7.28 (1H, dd, J1 = 8.2 Hz, J2 = 1.9 Hz, H20), 7.18 (3H, m, H14, H16, H25), 7.04 (2H, d, J = 8.5 Hz, H2, H6), 7.05 (1H, d, J = 8.2 Hz, H24), 6.92 (3H, m, H13, H17, H1), 6.84 (2H, d, J = 3.3 Hz, H3, H5), 6.12 (2H, s, H22), 3.74 (3H, s, H26), 2.91 (4H, br, H7, H10), 2.43 (4H, br, H8, H9); ^13^C NMR (500 MHz, DMSO-*d6*) δ (ppm): 152.00 (C5), 150.98 (C23), 147.79 (C21), 141.26 (C4), 132.87 (C15), 129.82 (C12), 122.51 (C19), 122.41–106.44 (C1, C2, C3, C6, C13, C14, C16, C17, C20, C24, C25), 102.56 (C22), 61.55 (C11), 55.33 (C26), 52.79 (C8, C9), 50.05 (C7, C10); IR (ATR, cm^−1^): 1328 and 1150 (ʋ S=O), 3247 and 1595 (ʋ N-H). Purity (HPLC): 97.0%; Anal. calcd. for C25H27N3O5S: C, 62.35; H, 5.65; N, 8.73; Found: C, 62.18; H, 5.67; N, 8.76. 

##### *N*-(4-((4-(2,3-Dichlorophenyl)piperazin-1-yl)methyl)phenyl) Benzo [d] [1,3] Dioxole-5-sulfonamide (**5f**)

*N*-Phenylpiperazine derivative **5f** was obtained in 55% yield as a yellowish solid, with a melting point of 163–166 °C. ^1^H NMR (500 MHz, DMSO-*d6*) δ (ppm): 7.19 (1H, m, H20), 7.27 (2H, m, H14, H16), 7.18 (2H, d, J = 1.9 Hz, H13, H17), 7.17 (1H, s, H25), 7.10 (1H, dd, J1 = 6.8, J2 = 2.8 Hz, H24), 7.04 (2H, d, J = 8.5 Hz, H1, H2), 7.00 (1H, d, J = 8.2 Hz, H3); ^13^C NMR (500 MHz, DMSO-*d6*) δ (ppm): 151.29 (C4), 147.85 (C23), 136.64 (C21), 133.78 (C15), 132.90 (C6), 132.69 (C12), 129.91 (C19), 128.54 (C5), 126.08–106.5 (C1, C2, C3, C13, C14, C16, C17, C20, C24, C25), 102.62 (C22), 61.43 (C11), 52.62 (C8, C9), 50.96 (C7, C10); IR (ATR, cm^−1^): 1335 and 1146 (ʋ S=O), 3108 and 1582 (ʋ N-H). Purity (HPLC): 97.0%; Anal. calcd. for C24H23Cl2N3O4S: C, 55.39; H, 4.45; N, 8.07; Found: C, 55.49; H, 4.44; N, 8.05. 

### 2.2. Molecular Modeling

Molecular docking studies were performed using Genetic Optimization for Ligand Docking (GOLD) *v*. 5.6 [35,36,37,38,39,40]. Crystallographic structures of the D_3_ and D_2_ receptors were selected from the Protein Data Bank (PDB; http://www.rcsb.org, accessed on 22 June 2022) protein database. The crystallographic structure with the 3PBL code (resolution 2.89 Å) in PDB was selected for the D_3_ receptor [41], whereas the chosen crystallographic structure of the D_2_ receptor was that with the 6CM4 code (2.86 Å) [42]. Such structures were the only structures available in the PDB for these subtypes of dopamine receptors when this work was done.

Based on the structure of the cocrystallized ligand with the D_2_ receptor (risperidone), the validation of the methodology that would be used for molecular docking studies was carried out. Risperidone was chosen because it is structurally similar to the compounds designed for this work.

Hydrogen atoms were added to the protein, and the location of the binding site was defined using the cocrystalized ligand (risperidone) and all amino acids 6 Å away from it as a reference.

To carry out both the redocking and further studies, risperidone and other proposed molecules were built in the ChemDraw program, and then the protonation state of the molecules was analyzed using the Percepta program. Subsequently, the equilibrium geometry was calculated by the semiempirical method PM6 (Parametric Method 6) for the lowest energy conformers, which were then used for docking [43].

Since the available crystallographic structure 6CM4 does not present water molecules, redocking of risperidone in the D_2_ receptor was performed in the absence of them. Redocking results were evaluated by root-mean-square deviation (RMSD) calculation (more details in the Appendix A).

For a punctual evaluation of the interaction profile presented by the compounds, we performed a protein–ligand interaction profile analysis (PLIP—https://plip-tool.biotec.tu-dresden.de/plip-web/plip/index, accessed on 4 August 2022) [44].

### 2.3. Binding and GTP-Shift

Membrane preparations of recombinant Chem-1 cells (ChemiscreenTM, Millipore, Burlington, MA, USA), transfected by a process using human cDNAs encoding the D_3_ isoform of the dopaminergic receptor, were used.

Test substances were solubilized in 100% DMSO (stock solution) and then serially diluted in water. Nonspecific binding was measured in the presence of 30 µM sulpiride (a selective antagonist of the central dopamine receptors: D_2_, D_3_, and D_4_). To evaluate the intrinsic efficacy, 50 mM Tris-HCl and 5 mM KCl buffer were used [45]. In this protocol, we used a medium containing a high concentration of divalent cations (MgCl_2_ 5 mM and CaCl_2_ 1.5 mM), which favors the binding of agonists to the receptor, or a medium with high concentrations of sodium and guanosine triphosphate (GTP) (154 mM NaCl and GTP 1 mM), which hinders the binding of agonists.

After incubation with 2.25–5 µg of protein and the radioligand [^3^H]-spiperone (0.5 nM) at 37 °C for 2 h, filtration was performed. Filters previously soaked in polyethyleneimine solution (PEI 0.5%) were used, and they were quickly washed with 3 × 4 mL of ice-cold 5 mM Tris-HCl (pH 7.4).

Finally, to evaluate the selectivity of the substances, classic binding to D_2_-like receptors was performed using rat striatal membranes. Adult male Wistar rats (2.5–3 months) were killed by decapitation, their brains were immediately removed on ice and the striatum was dissected and stored in liquid nitrogen until use. This procedure was approved by the Institutional Ethical Committee for Animal Care from the Federal University of Rio de Janeiro (CEUA no. 052/19; 30 April 2019). The striatum was homogenized in a motorized Potter-type apparatus with a Teflon piston at 4 °C at 20 volumes per gram of tissue in ice-cold 50 mM Tris-HCl buffer (pH 7.4) containing 8 mM MgCl_2_ and 5 mM ethylenediaminetetraacetic acid (EDTA). The resulting suspension was ultracentrifuged at 48,000× *g* at 4 °C for 20 min. The pellet was resuspended in 20 volumes of the same buffer and incubated at 37 °C for 10 min to remove endogenous neurotransmitters. This suspension was cooled and ultracentrifuged at 48,000× *g* for 20 min at 4 °C [46]. The final pellet was resuspended and stored in liquid nitrogen until use.

In a medium containing the antagonist radioligand [^3^H]-YM-09151-2 0.1 nM, 120 mM NaCl, 5 mM KCl, 5 mM MgCl_2_, 1.5 mM CaCl_2_, 1 mM EDTA and 50 mM Tris-HCl (pH 7.2 a 25 °C), 50 µg of mouse striatum membrane was incubated in the dark (sodium light) for 60 min at 37 °C. Nonspecific binding was estimated using sulpiride (30 µM).

The affinity of substances for D_3_R and D_2_-like receptors was evaluated through classical competition assays to determine the IC_50_ value. Data were analyzed by nonlinear regression using the GraphPad Prism^®^ program (version 5.00) and the “binding-one site competition” model to adjust the curve and calculate the mean inhibitory concentration (IC_50_). For D_2_R, the *K*_i_ value was calculated from the Cheng–Prusoff equation: *K*_i_ = IC_50_/[1 + (radioligand)/*K*_d_].

Analysis of intrinsic efficacy for D_3_R was performed through the displacement caused by sodium with GTP and the ratio of the IC_50_ obtained (in this condition) by the IC_50_ obtained in the medium with MgCl_2_ and CaCl_2_. As a control, an experiment was carried out with dopamine, the endogenous agonist of this receptor.

## 3. Results and Discussion

### 3.1. Chemistry

The target *N*-arylpiperazine derivatives (**5a**–**f**) were prepared in good yields through the nucleophilic substitution reaction of sulfonyl chlorides (**7a**–**b**) by 4-((4-phenylpiperazin-1-yl)methyl)anilines (**6a**–**c**) under microwave irradiation (Figure 1) [47]. Key intermediates (**6a**–**c**) were obtained from the metal-catalyzed reduction of the corresponding 1-(4-nitrobenzyl)-4-phenylpiperazine precursors (**8a**–**c**), as described in Figure 1 [32,48]. However, 4-nitrobenzylpiperazines (**8a**–**c**) were obtained from reductive amination with NaCNBH_3_ [49] of the imine intermediates formed in situ after the reaction of substituted N-phenylpiperazines (**9a**–**c**) and 4-nitrobenzaldehyde (**10**) (Figure 1). Finally, sulfonyl chlorides (**7a**–**b**) were obtained from electrophilic aromatic substitution of 3,4-methylenedioxybenzene (**7a**) and 3,4-methylenedioxytoluene (**7b**) following the sequence of reactions previously described [48,50].

The obtained *N*-phenylpiperazine derivatives (**5a**–**f**) were fully spectroscopically characterized, and their degree of purity was determined by reversed-phase HPLC analysis to be greater than 95%, which was considered adequate for the next step of investigating their binding affinities and efficacies for dopaminergic D_2_/D_3_ receptors.

### 3.2. Binding Affinity, Intrinsic Energy and Molecular Modeling Studies

The new *N*-arylpiperazine derivatives bind to both D_2_ and D_3_ receptors with similar micromolar affinities (Table 1). The presence of aromatic ring systems and basic nitrogen appears to make the *N*-phenylpiperazine scaffold the main molecular recognition element for the binding site of aminergic G-protein-coupled receptors (GPCRs) [51]. This hypothesis is qualitatively supported by the interaction profile of the compounds. The *N*-phenylpiperazine subunit occupies the region of the orthosteric site of the D_2_ and D_3_ receptors, both for aripiprazole and cariprazine (Figure 2A–D) and for the new derivatives, represented here by Compound **5a** (Figure 3A,B). According to docking studies, interactions at the orthosteric site are hydrophobic and involve amino acid residues such as serine, tryptophan and phenylalanine. However, by analysis of the specific molecular interactions, we can observe some important differences between them for each of the compounds. For example, in D_3_ receptors, both aripiprazole and cariprazine show interactions with three identical amino acid residues of the OBS, Asp110, Phe245 and Phe246, but 5a interacts only with Asp110, in addition to presenting a hydrophobic interaction with Ile183, similarly to cariprazine (Figure 4A–C). In D_2_ receptors, both the prototypes and **5a** present a salt bridge interaction with the Asp114 residue. However, in relation to the other interactions, in comparative terms, especially when we compare **5a** and aripiprazole, we perceive a profile that involves different residues. Aripiprazole interacts with residues such as Thr119, Trp386 and Phe390, and **5a** does not interact with any of them (Figure 5A–C).

The presence of substituents on the phenyl, linked to the piperazine ring, did not significantly modify the affinity for these receptors.

In addition to the orthosteric site, a secondary binding site (SBP) was identified in dopaminergic receptors, which seems to be related not only to the affinity that ligands may have for these receptors but also to the selectivity between the different subtypes [29,30].

According to the docking studies carried out here, both the prototype compounds (**2** and **3**) and the new derivatives showed hydrophobic interactions in the SBP of the D_2_ and D_3_ receptors. However, such interactions took place in different regions of the SBP, which can be explained by two structural characteristics. The first one is related to the different chemical subunits present in the analyzed compounds. In aripiprazole (**2**), we have a dihydroquinoline, whereas in cariprazine (**3**), a butyramide, and in the new derivatives (**5a**–**f**), a 1,3-benzodioxole subunit. Furthermore, the conformational differences caused by the different spacer groups also contribute to the interactions in different regions of the SBP. While in aripiprazole, the alkyl spacer has great conformational freedom, in the new derivatives, the interphenylene spacer is conformationally restricted (Figure 2A–D and Figure 3A,B). We can note that, in the SBP of D3, aripiprazole and cariprazine have two interactions in common (Leu89, Phe106), as does **5a** (Leu89, Glu90), but only the prototypes interact with Phe106 (Figure 4A–C). In D_2_ receptors, **5a** has no interaction in common with 2 and 3 in the SBP (Figure 5A–C). 

For both receptors, the interaction pattern of **2** and **3** appears to be more “hydrophobic”, while **5a** performs a greater number of hydrogen bonds. Since hydrophobic interactions are strongly related to the displacement of water molecules located around the hydrophobic groups of the ligand and the binding site, when they interact with each other [52], these differences could indicate a more entropic interaction profile for the prototypes, while compound **5a** would have a more enthalpic profile, as well as the other compounds analyzed. This difference in interaction profile could explain the difference in affinity at both receptors for the prototype compounds (**2** and **3**) versus the new derivatives.

As efficacy is as important as affinity for the therapeutic effect of a drug, we initially decided to estimate the intrinsic efficacy of the new compounds for the D_3_ receptor using a functional binding assay. The classic GTP-shift assay is based on the ternary complex model for GPCRs and has been validated for the D_3_ receptor [42]. This assay is based on the difference in affinity measured for agonists in the absence and presence of a high concentration of GTP (or a lower concentration of a nonhydrolyzable GTP analog) that is capable of destabilizing the ternary complex ARG (high affinity state of the receptor), which is formed by the agonist (A), the receptor (R) and the G protein (G).

Figure 6A shows the profiles of competition curves for the binding of [^3^H]-spiperone to D_3_ receptors using a full agonist (dopamine, Figure 6A) for validation purposes. In the presence of 154 mM NaCl and 1 mM GTP, the dopamine competition curve was shifted to the right, indicating a loss of affinity for D_3_ receptors. When an antagonist is used as a competitor, the addition of GTP has no effect, as was observed for derivative **5a**, since the competition curves in the absence and presence of GTP were superimposed, indicating that **5a** is a D_3_ receptor antagonist (Figure 6B). Very similar behavior was found for the other *N*-phenylpiperazine derivatives presenting R_1_ groups as methyl groups, such as **5c** and **5b**, which were classified as weak inverse agonists. However, in compounds **5d** and **5f**, where the steric hindrance promoted by the presence of a methyl group in the 1,3-benzodioxole ring was abolished, we found an intrinsic efficacy as a partial agonist. This kind of influence of a methyl group in the bioactive conformation of drugs and drug candidates is well discussed in a previous paper by our group [53].

These results are consistent with the structural requirements for D_3_ receptor antagonists, namely, an arylpiperazine subunit, a hydrogen-bonded donor/acceptor group, an aryl subunit and a suitable spacer [29,30].

The competition curves for the other compounds are presented in the Appendix A).

Although the new *N*-phenylpiperazine derivatives (**5a**–**f**) present similar binding affinities for D_2_ and D_3_, the best Compounds **5e** and **5f** combined the presence of an ortho-substituted phenyl group attached to the *N*-phenylpiperazine subunit with the absence of a methyl group in the 1,3-benzodioxole ring, which could favor the interactions of the vicinal sulfonamide group with both target receptors. Interestingly, Compounds **5e** and **5f** were able to modulate D_3_ receptors with different intrinsic efficacies as antagonists and partial agonists, respectively.

## 4. Conclusions

As concluding remarks, this work described a new series of substituted *N*-phenylpiperazines (**5a**–**f**) designed as interphenylene analogs of the antipsychotic drugs aripiprazole (**2**) and cariprazine (**3**), presenting a 1,3-benzodioxole group as a ligand of the secondary binding pocket of dopamine D_2_ and D_3_ receptors. The target compounds were synthesized in good yields by using classical methodologies, and their binding to both D_2_ and D_3_ receptor subtypes, as well as GTP shift studies, were performed. The best, Compounds **5e** and **5f**, presented affinity values of 0.1 and 0.2 μM (Ki for D_2_) and 0.2 and 0.2 μM (IC_50_ for D_3_), respectively, and distinct profiles of intrinsic efficacy. Docking studies revealed that Compounds **5a**–**f** present a different binding mode with dopamine D_2_ and D_3_ receptors, mainly as a consequence of the conformational restriction imposed on the flexible spacer groups of 2 and 3. Although the prototypes (2 and 3) and the new compounds are predicted to interact at the same binding sites, detailed analysis of the interaction profile indicate that the difference in affinity at both receptors for the prototype compounds versus the new derivatives could be related to differences in interactions with binding site residues. Taken together, these results indicated that the *N*-phenylpiperazine derivatives **5e** and **5f** are promising dual ligands of dopamine D_2_ and D_3_ receptor candidates for further studies in animal models of schizophrenia and drug addiction.

## Data Availability

Not applicable.

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
