# Peer review of "Design, Synthesis and Pharmacological Evaluation of Novel Conformationally Restricted N-arylpiperazine Derivatives Characterized as D2/D3 Receptor Ligands, Candidates for the Treatment of Neurodegenerative Diseases"

_biomolecules, 2022, doi:10.3390/biom12081112_

Round 1
Reviewer 1 Report
In the submitted manuscript authors synthesized a series of N-arylpiperazine derivatives composed of the arylpiperazine pharmacophore connected to the 1,3-benzodioxole moiety through the interphenylene spacer. The structural identity of intermediates and target compounds was confirmed by 1H NMR, 13C NMR, IR, and HPLC. Following compound characterization, authors performed a series of ligand docking studies and binding assays to determine the binding affinity of synthesized compounds for D2 and D3 dopamine receptors. Overall, the work resulted in novel compounds that were appropriately characterized. I recommend the manuscript for publication after the following questions are addressed:
1. Are high-resolution mass spectrometry (HRMS) data available for the synthesized compounds?
2. Authors provide dose-response competition curves for IC50 values (D3 dopamine receptor) in the Supplementary Information. However, no competition curves were provided for Ki values (D2 dopamine receptor). Please include these data in the revised submission.
3. Have the authors confirmed intrinsic efficacy at D3R (Table 1) using a functional assay, e.g. quantitative measurement of intracellular cAMP via time-resolved Förster resonance energy transfer (TR-FRET)?
Author Response
I want to thank the reviewer for the suggestions and commentaries about the work, which were very important for the improvement of the final form of our manuscript. We provide the response to the comments, as described next:
- Are high-resolution mass spectrometry (HRMS) data available for the synthesized compounds?
Authors reply: In fact, we don’t have high-resolution mass spectrometry (HRMS) data available for compounds 5a-f but, in order to confirm their purity, we have included elemental analysis data in the experimental section, which confirmed that obtained C, H, N results were within ±0.4% of calculated values.
- Authors provide dose-response competition curves for IC50 values (D3 dopamine receptor) in the Supplementary Information. However, no competition curves were provided for Ki values (D2 dopamine receptor). Please include these data in the revised submission.
Authors reply: I would like to thank the referee for the suggestion. In attention, we have added such competition curves for the D2R in the Supplementary material.
- Have the authors confirmed intrinsic efficacy at D3R (Table 1) using a functional assay, e.g. quantitative measurement of intracellular cAMP via time-resolved Förster resonance energy transfer (TR-FRET)?
Authors reply: We did not confirm the intrinsic efficacy at D3R by a second technique. Note that we have good experience in using classical functional binding assays for estimation of intrinsic efficacy at GPCRs, as detailed for the 5-HT1A receptor [Noël et al. (2014) Functional binding assays for estimation of the intrinsic efficacy of ligands at the 5-HT1A receptor: application for screening drug candidates. J. Pharmacol. Toxicol. Methods 70: 12-18] and the A2A receptor [Noël et al. (2016) Na+-shift binding assay for estimation of the intrinsic efficacy of ligands at the A2A adenosine receptor.J. Pharmacol. Toxicol. Methods 84:51-56].
Reviewer 2 Report
In this paper, da Silva Cunha et al report the design, synthesis and pharmacological evaluation of N-arylpiperazines, as D2/D3 ligands. The work consists of experimental and in silico parts.
The docking studies part is insufficiently explained and requires deeper analysis. Namely, the authors explain that "Docking studies revealed that Compounds 5a-f present a different binding mode with
dopamine D2 and D3 receptors, mainly as a consequence of the conformational restriction imposed on the flexible spacer groups of 2 and 3." This concerns the conformations of the novel compounds where a direct salt bridge with D3.32 is not recognized. This can be resolved by creating a series of conformers and doing consecutive docking into the receptor. The authors should try to carry this in Autodock/ Autodock Vina as control software. The ligand-receptor interactions should be listed in detail and used to explain the trends in binding affinities and correlations to experimental measurements.
Author Response
I want to thank the reviewer for the suggestions and commentaries about the work, which were very important for the improvement of the final form of our manuscript. We provide the response to the comments, as described next:
The docking studies part is insufficiently explained and requires deeper analysis. Namely, the authors explain that "Docking studies revealed that Compounds 5a-f present a different binding mode with dopamine D2 and D3 receptors, mainly as a consequence of the conformational restriction imposed on the flexible spacer groups of 2 and 3." This concerns the conformations of the novel compounds where a direct salt bridge with D3.32 is not recognized. This can be resolved by creating a series of conformers and doing consecutive docking into the receptor. The authors should try to carry this in Autodock/ Autodock Vina as control software. The ligand-receptor interactions should be listed in detail and used to explain the trends in binding affinities and correlations to experimental measurements.
Authors reply: In fact, there are many excellent options for molecular docking software, which offer different combinations of scoring functions and conformational search algorithms, all of them with some advantages and disadvantages. Autodock uses a Lamarckian Genetic Algorithm (Morris et al., J. Comput. Chem. 30 (2009), 2785-2791), which is based on a random/stochastic search. During a docking procedure with this type of algorithm, random modifications are made in the conformation and orientation of a ligand inside the binding site in the search of the best fitted pose. In this sense, using different conformations of a ligand and doing consecutive docking with them would be of little effect, since each one would be randomly modified at the beginning of the docking run.
Our choice of GOLD was based on our previous experience with this docking software and on the successful risperidone redocking study with an excellent RMSD value for the ChemPLP scoring function, as described in the manuscript. It is not clear to us why we should use Autodock/ Autodock Vina as control software as suggested by the referee. The available literature indicates that Autodock Vina and GOLD docking performances are good, and they do not present a significantly superior performance when compared to each other. For example, it was shown that GOLD (ChemPLP scoring function) and Autodock Vina both exhibited exactly the same docking success rate (82.4%), using the RMSD between the experimentally observed ligand conformation and the best pose predicted by the docking program as a criterion, with a data set comprised 34 protein–ligand complexes, including 18 diverse protein families, and ligands with distinct physicochemical, structural properties and a wide range of ligand rotatable bonds and formal charges (de Magalhães et al., Information Sci. 289 (2014) 206–224).
In order to better discuss the interactions observed qualitatively by molecular docking studies, we performed a Protein-Ligand Interaction Profiler (PLIP) analysis for the prototype compounds (aripiprazole and cariprazine) and for the new derivatives. We have included in the text the discussion of the analysis together with the Figures and Tables corresponding to the interactions of the prototypes and the compound used as an example (5a). Other Figures and Tables were added to the Supplementary Material.
The following text was added in the discussion section:
“According to docking studies, interactions at the orthosteric site are hydrophobic and involve amino acid residues such as serine, tryptophan and phenylalanine. However, by analysis of the specific molecular interactions, we can observe some differences between them for each of the compounds. For example, in D3 receptors, both aripiprazole and cariprazine show interactions with 3 identical amino acid residues of the OBS, Asp110, Phe245 and Phe246, but 5a interacts only with Asp110, in addition to presenting a hydrophobic interaction with Ile183, such as cariprazine. In D2 receptors, both the prototypes and 5a present a salt bridge interaction with the Asp114 residue. However, in relation to the other interactions, in comparative terms, especially when we compare 5a and aripiprazole, we perceive a profile that involves different residues. Aripiprazole interacts with residues such as Thr119, Trp386 and Phe390, and 5a does not interact with any of them.”
“While in aripiprazole the alkyl spacer has great conformational freedom, in the new derivatives the interphenylene spacer is conformationally restricted (Figure 2A-D and Figure 3A-B). We can note that in the SBP of D3, aripiprazole and cariprazine have two interactions in common (Leu89, Phe106), as does 5a (Leu89, Glu90), but only the prototypes interact with Phe106. In D2 receptors, 5a has no interaction in common with 2 and 3 in the SBP.
For both receptors, the interaction pattern of 2 and 3 appears to be more “hydrophobic”, while 5a performs a greater number of hydrogen bonds. Since hydrophobic interactions are strongly related to the displacement of water molecules located around the hydrophobic groups of the ligand and the binding site, when they interact with each other (Bissantz, C.; Kuhn, B.; Stahl, M. J. Med. Chem. 2010, 53, 5061), these differences could indicate a more entropic interaction profile for the prototypes, while compound 5a would have a more enthalpic profile, as well as the other compounds analyzed. This could explain the difference in affinity at both receptors for the prototype compounds (2 and 3) versus the new derivatives.”
Reviewer 3 Report
The Authors presented an interesting paper demonstrating novel compounds designed to treat neurodegenerative disorders. Although this theme is of great importance and constantly gains much attention, some issues need to be discussed, and eventually corrected.
1. Neurodegenerative disorders are quite complex and dopamine receptors may play a contrary role. Moreover, the main subtype of the dopamine receptor involved in the disease also vary. Therefore, the Authors should indicate the type of the neurodegenerative state for which they designed and analyzed the compounds mentioned in the text.
2. The aim of the study needs to be highlighted. In fact, there is no information regarding the reason for which antipsychotic drugs were taken as the basis of the study. There should be some information that antipsychotic treatment is effective in the psychotic and behavioural disturbances of neurodegenerative disorders.
3. In line with this, also a small introduction of a safety profile of dopamine receptor ligands (in terms of their addictive behavior) should be given.
4. Im quite confused with the fact that the synthesized compounds were solubilized in 100%!! DMSO. What was the final concentration of the DMSO in diluted substances since it is known that high concentration (>10% or even less) of DMSO is harmful for cells.
5. Please correct in the methodology section the use of mouse or rat brain homogenates.
6. The Authors provided that the new compounds "bind to both D2 and D3 receptors with similar affinities", which should be given in the table 1. However, the Authors determined affinities in vivo only for DRD2, not for DRD3 for which intrinsic activity (i.e., agonist/antagonist/partial agonist/inverse agonist) was determined. Please correct this sentence. Otherwise, rewrite the text.
7. In the table 1 the Authors should provide data for risperidone and sulpiride, which were used for the analysis and calculations.
8. Figure 4, showing the GTP gamma S binding, should contain both the control compound and the tested compound on the same graph, with the IC50 value provided
9. Since the D2 receptor is crucial in both the psychotic and neurodegenerative diseases and disorders, please state the reason for which only the intrinsic activity at D3 receptor was evaluated.
10. Similarly, the affinities do not provide any information on the antagonist or agonist profile of the compound. Therefore, I was wondering whether the Authors could improve the manuscript by providing the Emax value for D2 receptors?
11. There is little discussion and introduction regarding the usefulness of the compounds in dopamine-based disorders and the relation of D2 and D3 receptors with the neurodegeneration state.
12. In the inclusion section, the Authors stated that the designed and synthesized compounds are great candidates for animal models of schizophrenia and drug addiction. I'm confused. These two "disorders" are not neurodegenerative ones. Therefore, the conclusion should be definitely changed.
Author Response
I want to thank the reviewer for the suggestions and commentaries about the work, which were very important for the improvement of the final form of our manuscript. We provide the response to the comments, as described next:
- Neurodegenerative disorders are quite complex and dopamine receptors may play a contrary role. Moreover, the main subtype of the dopamine receptor involved in the disease also vary. Therefore, the Authors should indicate the type of the neurodegenerative state for which they designed and analyzed the compounds mentioned in the text.
Authors reply: In fact, the design of novel compounds was not mainly motivated by a specific disease but for the possibility of identifying dual modulators of dopamine D2 and D3 receptors, presenting different intrinsic activity. So, depending on the prevalence of these isoforms in different regions of CNS, with a key role in the pathogenesis of each neurodegenerative disease, and the relative efficacy of compounds in modulate it, we can anticipate that distinct aryl-sulfonamide-N-phenylpiperazine derivatives (5a-f) could present better profiles for different diseases.
- The aim of the study needs to be highlighted. In fact, there is no information regarding the reason for which antipsychotic drugs were taken as the basis of the study. There should be some information that antipsychotic treatment is effective in the psychotic and behavioural disturbances of neurodegenerative disorders.
Authors reply: Its affirmation is not completely true, because, for example compound (4) that was also used in the design of target compounds is not an antipsychotic drug, but as an antagonist of D3 receptors it was designed for treatment of drug addiction. So, the main idea was promoting the hybridization of these structures and evaluate how the conformation restriction promoted by methylation and the presence of an interphenylene spacer, could affect the intrinsic efficacy.
3. In line with this, also a small introduction of a safety profile of dopamine receptor ligands (in terms of their addictive behavior) should be given.
Authors reply: To meet the suggestion, we have introduced a new reference, i.e. [29], in the introduction describing the importance and safety of dopamine ligands useful for drug addiction.
“Maramai, S.; Gemma, S.; Brogi, S.; Campiani, G.; Butini, S.; Stark, H.; Brindisi, M. Dopamine D3 Receptor Antagonists as Potential Therapeutics for the Treatment of Neurological Diseases. Frontiers in Neuroscience 2016, 10, article 451.”
- Im quite confused with the fact that the synthesized compounds were solubilized in 100%!! DMSO. What was the final concentration of the DMSO in diluted substances since it is known that high concentration (>10% or even less) of DMSO is harmful for cells.
Authors reply: The stock solutions (10 mM) were obtained by solubilizing the compounds in 100% DMSO and then serially diluted in water just before use, with no more than 0.1 % DMSO in the final incubation medium, a concentration that has no effect on the binding. In order to clarify this point, we now added the following comment after the first sentence of the second paragraph of section 2.3: “Test substances were solubilized in 100% DMSO (stock solution) and then serially diluted in water. At the final concentration used (below 0.1%), DMSO did not affect our binding assays.”
- Please correct in the methodology section the use of mouse or rat brain homogenates.
Authors reply: Thank you for your comment. We corrected the text of the fifth paragraph of section 2.3.
“In a medium containing the antagonist radioligand [3H]-YM-09151-2 0.1 nM, 120 mM NaCl, 5 mM KCl, 5 mM MgCl2, 1.5 mM CaCl2, 1 mM EDTA and 50 mM Tris-HCl (pH 7.2 a 25 °C), 50 µg of rat striatum membrane was incubated in the dark (sodium light) for 60 minutes at 37 °C. Nonspecific binding was estimated using sulpiride (30 µM).”
This information was already available in the methodology since we indicated that “Adult male Wistar rats (2.5-3 months) were killed by decapitation, their brains were immediately removed ….”
- The Authors provided that the new compounds "bind to both D2 and D3 receptors with similar affinities", which should be given in the table 1. However, the Authors determined affinities in vivo only for DRD2, not for DRD3 for which intrinsic activity (i.e., agonist/antagonist/partial agonist/inverse agonist) was determined. Please correct this sentence. Otherwise, rewrite the text.
Authors reply: We did not perform any in vivo study. The apparent affinities (IC50 values) for DRD2 and DRD3 receptors were obtained in vitro by competition radioligand binding studies performed with membrane preparation from either rat brain (DRD2) or recombinant Chem-1 cells expressing the human D3 isoform (DRD3). For the D2RD, we corrected the IC50 values by using the Cheng-Prusoff equation in order to indicate the Ki values in the table. For the DRD3, we decided to indicate the IC50 values obtained in the two different media, in order to clarify how the Na-shift was calculated for estimating the intrinsic efficacy, something less classical for the general reader. Note that in our experimental assays, the radioligand concentration is always below its Kd value so that the IC50 values are good estimates of the Ki values (less than two-fold correction).
We now added the following comment to the first sentence of paragraph 3.2. in order to clarify this point without entering too many unnecessary details: “The new N-arylpiperazine derivatives bind to both D2 and D3 receptors with similar micromolar affinities, as indicated in Table 1 where the Ki values (D2R) or IC50 values (D3R) are reported. Note that in our conditions, the Ki values are not much different from the IC50 values since we used radioligand concentrations lower than their Kd values.”
- In the table 1 the Authors should provide data for risperidone and sulpiride, which were used for the analysis and calculations.
Authors reply: I understand the question pointed by referee, however risperidone was only used to validate redocking step, not being used in the comparative analysis of binding mode of our target N-phenylpiperazine derivatives. For this reason, risperidone was not used in our binding experiments. Additionally, sulpiride was only used at a high concentration to inhibit all the radioligand binding to the receptors, as reported in the literature.
Figure 4, showing the GTP gamma S binding, should contain both the control compound and the tested compound on the same graph, with the IC50 value provided
Authors reply: We did not perform GTP gamma S binding. If we put the two drugs (4 curves) on the same graph, the competition curve with dopamine in the presence of 154 mM NaCl and 1 mM GTP (blue curve) would be superimposed to the two competition curves with 5a. For this reason, we did prefer to show the two drugs in separated graphs in order to better observe the presence or not of a GTP shift, that was the aim of this experiment. Note that we were cautious using the same scales and range of concentration in the two graphs to avoid any bias.
9. Since the D2 receptor is crucial in both the psychotic and neurodegenerative diseases and disorders, please state the reason for which only the intrinsic activity at D3 receptor was evaluated.
Authors reply: The question is easy to answer. As there are a lot of drugs and drugs candidates that are useful to treat neurodegenerative diseases by acting as D2R modulators and, when these compounds present a substituted N-phenylpiperazine subunit they are in most of cases antagonists (such as is the case of compounds 5a-f), we decided to focus the investigation of the intrinsic efficacy of our novel compounds in D3 receptors. This decision was also supported by the fact that nowadays D3 receptors are being considered the most important subtype of dopaminergic receptors for discovery of novel drug candidates for neurodegenerative diseases, due to the lower number of adverse effects, as indicated by reference:
“Maramai, S.; Gemma, S.; Brogi, S.; Campiani, G.; Butini, S.; Stark, H.; Brindisi, M. Dopamine D3 Receptor Antagonists as Potential Therapeutics for the Treatment of Neurological Diseases. Frontiers in Neuroscience 2016, 10, article 451.”
- Similarly, the affinities do not provide any information on the antagonist or agonist profile of the compound. Therefore, I was wondering whether the Authors could improve the manuscript by providing the Emax value for D2 receptors?
Authors reply: We did not estimate effect, neither for the D3R nor the D2R. For the D3R we used a functional binding assay for estimating the intrinsic efficacy of our compounds.
11. There is little discussion and introduction regarding the usefulness of the compounds in dopamine-based disorders and the relation of D2 and D3 receptors with the neurodegeneration state.
Authors reply: Regarding the description of compounds that act as D2/D3 modulators useful for treatment of neurodegenerative diseases, the second and third paragraph of second page, describe objectively the most important information:
“The importance of dopaminergic pathways and receptor modulators in the control of neurodegenerative diseases led to the development of drugs such as the classical typical antipsychotic haloperidol (1), a D2 receptor antagonist (Ki = 0.89 nM) [19,20] and the atypical antipsychotics aripiprazole (2) (Ki D2 = 0.34 nM; Ki D3 = 0.8 nM) and cariprazine (3) (Ki D3 = 0.085 nM; Ki D2 = 0.49 nM), as partial agonists of D2 and D3 receptors [21-26], approved for the treatment of schizophrenia and bipolar disorder (Figure 1).”
“Other analog N-phenylpiperazine compounds, now showing intrinsic efficacy as antagonists, have also been developed to act in the treatment of dependence and drug addiction [27,28] with adequate safety profile [29].”
On the other hand, a detailed description of the mechanisms associated with the effect of D2 and D3 receptors is completely out of the scope of this work.
12. In the inclusion section, the Authors stated that the designed and synthesized compounds are great candidates for animal models of schizophrenia and drug addiction. I'm confused. These two "disorders" are not neurodegenerative ones. Therefore, the conclusion should be definitely changed.
Authors reply: In fact, in the past there has always been a lot of controversy over whether or not schizophrenia is a neurodegenerative disease, but recent evidence confirms its nature:
“Stone, W. S.; Phillips, M. R.; Yang, L. H.; Kegeles, L. S.; Susser, E. S.; Lieberman, J. A. Neurodegenerative model of schizophrenia: Growing evidence to support a revisit. Schizophrenia Research 2022, 243, 154-162.”
A very similar profile could be evidence for drug addiction, once the chronic use of certain drugs, e,g. heroin, could result in the development of a neurodegenerative process:
“Mohammad Ahmadi Soleimani, S.; Ekhtiari, H.; Cadet, J. L. Drug-induced neurotoxicity in addiction medicine. Neuroscience for Addiction Medicine: From Prevention to Rehabilitation - Constructs and Drugs, 19–41 (2016).
“Zhu, M., Xu, Y., Wang, H., Shen, Z., Xie, Z., Chen, F., Gao, Y.; Chen, X.; Zhang, Y.; Wu, Q.; Li, X.; Yu, J.; Luo, H.; Wang, K. Heroin Abuse Results in Shifted RNA Expression to Neurodegenerative Diseases and Attenuation of TNFα Signaling Pathway. Scientific Reports 2018, 8(1).”
Round 2
Reviewer 2 Report
The responses provided by the authors are satisfactory. I endorse this publication.
Reviewer 3 Report
The paper is now accepted